# INCREMENTAL LEARNING WITH TASK-SPECIFIC ADAPTERS

## ABSTRACT

Incremental learning aims to continuously acquire new knowledge while preserving previously learned information. Existing literature primarily focuses on improving model stability, often at the cost of plasticity, to prevent the forgetting of earlier tasks. In this paper, we argue that inter-task differences are the primary driver of catastrophic forgetting. To address this challenge, we propose a novel network architecture compromising two distinct components: one dedicated to learning invariant features shared across tasks and another for capturing task-specific details. Specifically, we repurpose adapters, originally introduced for parameter-efficient fine-tuning, as feature modifiers to capture task-specific details, while the backbone network focuses on learning invariant features. Unlike prior approaches that keep the backbone frozen and only fine-tune adapters, we co-train both the backbone network and adapters, employing an additional regularization term that encourages the backbone to learn shared features. Our approach integrates seamlessly with established methods, such as Learning without Forgetting (LwF). Extensive experiments on CIFAR-100 and ImageNet datasets demonstrate that our adapter-based methods consistently outperform non-adapter counterparts across diverse learning scenarios, including various task orders and data scales. Our approach improves both plasticity and stability, effectively addressing the stability-plasticity dilemma.

## 1 INTRODUCTION

Real-world data often arrive in a sequential manner, in batches, or through periodic updates. These data dynamics are further complicated by constraints such as limited storage capacity and privacy considerations. In such contexts, employing concurrent or multitask learning, where models are trained on a single, static, large dataset, can be costly or impractical. Incremental learning (IL), also known as lifelong learning, is specifically designed to handle these dynamic environments. This paradigm allows a model to continuously learn and update its knowledge from new data without retraining from scratch. Unlike traditional methods that require access to the entire dataset for training, incremental learning enables the model to adapt to new tasks over time using only the data provided for each new task while retaining knowledge from previous tasks.

Feature extraction and fine-tuning are two common approaches for learning new tasks without accessing previous training data. In feature extraction, the weights of a pre-trained model are kept fixed, and the outputs of the top layer are used as features for the new task (Donahue et al., 2014; Belouadah & Popescu, 2018). While this approach helps maintain existing knowledge, it may struggle to effectively learn new tasks. Fine-tuning improves upon feature extraction by updating the model weights for the new task. A low learning rate is typically used to ensure the model retains the structure and knowledge from previous tasks (Girshick et al., 2014). While this approach enables better adaptation to new tasks, it carries the risk of *catastrophic forgetting*, where performance on previously learned tasks deteriorates rapidly (Goodfellow et al., 2013; McCloskey & Cohen, 1989). Forgetting occurs when the task-specific weights of the previous tasks are altered to accommodate new tasks. These methods may not fully address the challenges of incremental learning, which requires balancing the preservation of existing knowledge with the acquisition of new skills.

An ideal incremental learning algorithm should at least possess the following key properties: efficient memory usage and stability-plasticity balance. First, the algorithm should optimize memory usage, preserving no or only essential data points from previous tasks. This may reflect realistic scenarios

where data might be transient or unavailable due to privacy or memory constraints. Second, it should maintain a balance between stability (retaining past knowledge) and plasticity (learning new information), avoiding catastrophic forgetting (losing old knowledge when learning new data). Balancing stability and plasticity is a fundamental challenge in incremental learning, often referred to as the stability-plasticity dilemma (Mermillod et al., 2013).

Existing research in incremental learning primarily focuses on mitigating catastrophic forgetting by improving stability at the expense of plasticity. One popular approach is to directly regularize weight changes (Kirkpatrick et al., 2017; Zenke et al., 2017; Aljundi et al., 2018) or to limit divergence in output predictions, between the old and new models (Li & Hoiem, 2017; Dhar et al., 2019). This strategy can be extended by incorporating data retained from previous tasks, which further prevents forgetting by replaying some of the prior knowledge (Chaudhry et al., 2018; Rebuffi et al., 2017; Zhang et al., 2020b). However, enhancing stability often hurts plasticity. Few works simultaneously improve stability and plasticity.

**Contribution** In this paper, we argue that the inter-task differences contribute to catastrophic forgetting and propose to model these differences. Specifically, we propose a network design consisting of two blocks: a backbone network for learning invariant features across all tasks and many small networks for modeling task-specific knowledge. Specifically, we use adapters, originally introduced for parameter-efficient fine-tuning of large language models (Houlsby et al., 2019), as these small networks to capture the task-specific information. Our approach differs from the conventional use of adapters, which are typically added to a frozen network to attain comparable performance to full fine-tuning. Instead, we re-purpose adapters as feature modifiers and train them together. This strategy enables the adapters to encapsulate task-specific information in the layers closer to the output, while squeezing task-invariant knowledge into layers nearer the input. Our approach can be integrated with many existing methods such as EWC and LwF. Our empirical results demonstrate that various adapter-assisted methods consistently outperform non-adapter counterparts along the learning process, ranging from regularization-based to rehearsal-based approaches, across both CIFAR-100 and ImageNet datasets. The advantage remains robust among dataset choices, task scales, and task orderings. Our approach improves both plasticity and stability, eliminating the stability-plasticity dilemma.

## 2 RELATED WORK

This section reviews works that are closely related to ours.

**Multi-task Learning** Multi-task learning involves training on all tasks simultaneously, leveraging shared network parameters to exploit inter-task commonalities, in contrast to incremental learning (Caruana, 1997; Ruder, 2017). However, multi-task learning can be costly or impractical due to the substantial computational burden of training on large datasets, alongside challenges such as limited storage capacity and privacy concerns (Kendall et al., 2018). These limitations highlight the importance of incremental learning as a critical area of research.

**Incremental Learning** Incremental learning involves learning tasks sequentially. However, this approach is susceptible to catastrophic forgetting, where learning new tasks can overwrite previously acquired knowledge (McCloskey & Cohen, 1989). To address this challenge, three primary strategies are commonly employed: regularization-based, rehearsal-based, and parameter-isolation methods. Regularization-based methods mitigate forgetting by regularizing differences in weights or output predictions between the old and new models (Kirkpatrick et al., 2017; Zenke et al., 2017; Aljundi et al., 2018; Li & Hoiem, 2017; Dhar et al., 2019; Joseph et al., 2022). Rehearsal-based methods preserve prior knowledge by retaining instances from previous tasks and training on a combined dataset that includes these instances alongside data from the new task. This retained data may consist of exemplar images (Rebuffi et al., 2017; Chaudhry et al., 2018), publicly available external datasets (Lee et al., 2019; Zhang et al., 2020b), or synthetic data generated by generative models (Shin et al., 2017; Kemker & Kanan, 2017; He et al., 2018). Parameter-isolation methods improve stability by freezing parameters that are critical to previously learned tasks (Mallya & Lazebnik, 2018; Serra et al., 2018).

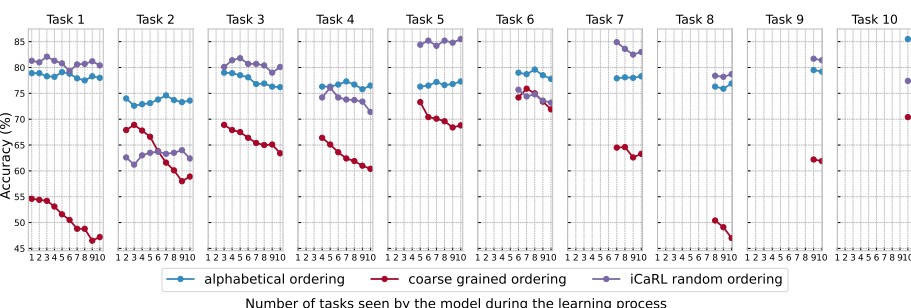

Figure 1: The model's accuracy using LwF for incremental learning is evaluated on three different CIFAR-100 task orderings: the standard alphabetical category ordering, the coarse grained ordering, and a random ordering with the fixed seed 1993 used by iCaRL (Rebuffi et al., 2017). The coarse grained ordering has more inter-task diversity.

While each of the aforementioned strategies has its strengths, they also present notable limitations. Regularization-based and parameter-isolation approaches improve model stability but often compromise plasticity, and rehearsal-based methods encounter practical challenges, including storage constraints, privacy concerns, and scalability issues. Recent research has proposed adapter-based methods to address catastrophic forgetting(Rajasegaran et al., 2020; Bhat et al., 2023; Pham et al., 2021; Wang et al., 2022; Liang & Li, 2024; Zhang et al., 2020a). Some of these works incorporate adapter-like subnets. However, they are limited in several ways: 1) freezing the backbone (Liang & Li, 2024; Zhang et al., 2020a) can negatively impact performance as the shared information can not be learned effectively; 2) relying on custom, complex loss functions (Liang & Li, 2024; Bhat et al., 2023) reduces compatibility with new algorithms and limits broader applicability; and 3) none incorporates regularization- or prediction-based approaches. By focusing exclusively on network architecture, these methods fail to leverage valuable insights from robust yet foundational baselines such as Learning without Forgetting (Li & Hoiem, 2017). To address these shortcomings, we propose leveraging adapters to develop a lightweight and compatible solution that combines strong performance with insights from both adapter design and algorithmic principles.

**Fine-tuning with adapters**    Another line of research explores the potential of adapters for transferring knowledge to downstream tasks, particularly within large language models. Adapters are compact modules inserted between the layers of a pre-trained large model (Houlsby et al., 2019). These modules are typically fine-tuned while the original network remains frozen. Fine-tuning with adapters achieves performance comparable to full fine-tuning across diverse tasks (Li & Liang, 2021). Moreover, adapters facilitate rapid adaptation to new tasks without catastrophic forgetting (Pfeiffer et al., 2020a), addressing challenges in multi-domain (Chronopoulou et al., 2023; Asai et al., 2022) and multilingual settings (Pfeiffer et al., 2020b).

## 3 INCREMENTAL LEARNING WITH TASK-SPECIFIC ADAPTERS

This section highlights a key limitation of current knowledge distillation methods for incremental learning and introduces adapters as feature modifiers to model inter-task differences.

### 3.1 WHAT IS MISSING IN REGULARIZATION-BASED METHODS?

A key limitation of regularization-based methods is their susceptibility to the stability-plasticity dilemma. Specifically, anchoring the model to its performance on prior tasks limits its plasticity in learning new tasks, while relaxing the regularization leads to catastrophic forgetting. This issue becomes more pronounced with greater inter-task diversity in the incremental learning problem. Variations in inter-task diversity can be introduced through different task orderings.

Figure 1 examines the impact of task orderings on the classification performance of the LwF algorithm applied to CIFAR-100. In this scenario, the model is incrementally trained with 10 classes per task over a total of 10 tasks. Three different task orderings are evaluated: an alphabetical class ordering, a

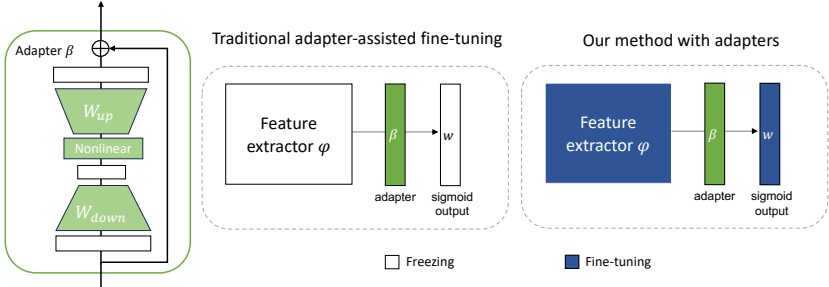

Figure 2: Architecture of the adapter and a comparison highlighting the distinctions in its implementation between traditional fine-tuning and our method. Left: an adapter consists of the down-projection, the nonlinear transformation, up-projection, and skip-connection. Right: The key difference between traditional use of adapter and ours is that we allow adapters to be co-trained with the entire network when learning a new task.

random ordering with a fixed seed commonly used by iCaRL and other methods (Rebuffi et al., 2017; Zhang et al., 2020b), and a coarse grained ordering that groups similar classes within each task based on CIFAR-100's 20 coarse categories.

The coarse grained ordering has greater inter-task diversity, providing a way to assess how increased inter-task differences affect the incremental learning algorithm. Notably, when classes are learned in a coarse-grained ordering, there is a significant increase in both forgetting and accuracy loss for each task compared to other orderings. This increased forgetting and accuracy decline can be attributed to the stability-plasticity dilemma of the LwF algorithm when confronted with greater inter-task diversity.

Therefore, there is a need to develop new strategies that can eliminate the stability-plasticity dilemma of regularization-based approaches, enabling them to learn new tasks without forgetting.

### 3.2 INTRODUCING ADAPTERS

In this section, we propose a network architecture comprising two distinct components: a backbone network for learning invariant features shared across all tasks and multiple lightweight adapters for capturing task-specific information. These adapters enhance the plasticity of the architecture, enabling it to adapt to new tasks, while the backbone network maintains stability by focusing on shared and invariant features. Unlike existing approaches that train adapters while keeping the backbone frozen, we use adapters as task-specific feature modifiers and co-train them alongside the backbone network. As the model trains on additional tasks and samples, the backbone network refines its ability to learn invariant features, further enhancing stability. This architectural design improves both stability and plasticity, effectively eliminating the stability-plasticity dilemma.

The adapters are positioned between the backbone feature extractor, denoted as $\varphi$, and the label predictor layer, serving as task-specific feature modifiers $\beta^t$ for each task $t$. As illustrated in Figure 2, the adapters adopt a conventional bottleneck structure. Starting with an initial dimension $d$ and a bottleneck width $b$, we design the down-projection layer to reduce dimensionality from $d$ to $b$ using a fully-connected neural network with a weight matrix $\mathbf{W}_{d \times b}$ and a non-linear activation function $g$, expressed as

$$\mathtt{Down}_{d \to b}(\mathbf{x}) = g\left(\mathbf{x}\mathbf{W}_{d \times b}\right).$$

Similarly, the up-projection layer is defined as

$$\mathtt{Up}_{b \to d}(\mathbf{x}) = g\left(\mathbf{x}\mathbf{W}_{b \times d}\right).$$

Our adapters consist of both a down-projection and an up-projection step and are connected to the output via a skip-connection (He et al., 2021). This bottleneck design allows the adapter to utilize both backbone features from $\varphi(x)$ and the modified features processed through the down- and up-projection layers:

$$\beta^t(\varphi(x)) = \varphi(x) + \mathtt{Up}_{b \to d}^t\left(\mathtt{Down}_{d \to b}^t\left(\varphi(x)\right)\right).$$

Our adapter module incorporates three key features specifically designed for incremental learning: (i) a compact design with a small number of parameters; (ii) compatibility with and enhancements over existing methods; and (iii) the simultaneous updating of feature extractor layers and adapters. The compact design addresses memory constraints, while the latter two features jointly improve stability and plasticity.

For (i), our method leverages adapters to control the growth rate of the overall model size when accommodating additional tasks. When applied to large backbone networks, the parameters introduced by the adapters are negligible, making our approach well-suited for models with memory constraints. This contrasts with strategies that rely on dynamically expanding networks (Yoon et al., 2017; Yan et al., 2021).

For (ii) and (iii), we address the stability-plasticity dilemma through a novel two-block design and their co-training. Specifically, adapters are trained to enhance plasticity, while the continuous training of backbone networks focuses on improving invariant feature learning, thereby further enhancing stability. Such a distinction is particularly challenging in traditional, non-adapter architectures, as their holistic design inherently exacerbates the stability-plasticity dilemma.

Lastly, to enforce that the backbone network learns invariant features and thus the adapters learn task-specific information, we develop method-specific regularization techniques. For the prediction-regularized methods such as LwF, we impose a knowledge distillation loss on the backbone, encouraging the backbone is similar before and after learning each new task. For the weight-regularized methods such as EWC, we free adapters from regularization. The approaches and designs are discussed in the following section in detail.

### 3.2.1 ENFORCING INVARIANT FEATURE LEARNING IN THE BACKBONE

We perform necessary modifications to the incremental learning methods so that the adapters can fit them and perform well. This section presents the adjustments for such integrations with different regularization methods.

**Prediction-regularized Methods**   The prediction-regularized methods attempt to address the stability-plasticity dilemma through model distillation instead of weight control. Algorithms such as LwF and Learning without Memorizing (LWM) (Dhar et al., 2019) fall into this category. Additional to the task loss, the model outputs at task $t$ are regularized with the model outputs at all tasks $t'$ where $1 \leq t' < t$, i.e. a distillation loss:

$$\mathcal{L}^t = \ell^t(\theta) + \lambda_{\text{distill}} R_{\text{distill}}^t$$

$$= \ell^t(\theta) + \lambda_{\text{distill}} \sum_{t'=1}^{t-1} M\left(\varphi^{t'}(x), \varphi^t(x)\right)$$

where $M$ is a metric that quantifies the similarities between the adapter outputs, such as the cosine similarity or cross entropy[1], and $\lambda_{\text{distill}}$ is a hyperparameter

This regularization method loosens the stability restriction of the network by distilling instead of direct controlling model parameters. It allows the model to search for an optimal solution in a larger parameter space. However, while we anticipate the adapter and the backbone model to capture the task-specific and task-invariant information respectively, this distillation constraint neither gives the adapters absolute freedom to the parameter search space nor poses a strong restriction for backbone task-invariance. The former cannot be done since adapters are involved in the forward pass and the computation of the distillation loss, and is arguably not a big deal due to that the restriction is already-loosen compared to the weight-regularized methods. Addressing the latter problem, we introduce an additional backbone regularization:

$$R_\varphi^t = \sum_{t'=1}^{t-1} M\left(\text{Linear}_{d \times c}(\varphi^{t'}(x)), \text{Linear}_{d \times c}(\varphi^t(x))\right),$$

where $c \leq d$ is a dimension we reduce to. In practical, we choose $c$ to be the number of classes of each task, as intuitively we make this regularization implicitly a direct distillation on backbones.

---

[1]Following previous works, we use the cross entropy loss as the metric.

Hence, to learn a new task $t$, we define a loss function that includes the task loss, the distillation term, and the backbone regularizer to align backbones across tasks to improve stability:

$$\mathcal{L}_t = \ell_t(\theta) + \lambda_{\text{distill}} R_{\text{distill}}^t + \lambda_\varphi R_\varphi^t, \tag{1}$$

where $\lambda_\varphi$ is the hyperparameter for the backbone regularizer. The parameter $\lambda_{\text{distill}}$ aims to balance retaining prior knowledge with adapting to new tasks during the incremental learning process, while $\lambda_\varphi$ controls the direction regularization to the backbone across tasks. We apply this adapter regularization on LwF as it is a prediction-regularized method and, in our experience, it is among the most effective methods for incremental learning tasks.

**Weight-regularized Methods** The weight-regularized methods control and regularize the weights of each task. Taking Elastic Weight Consolidation (EWC) (Kirkpatrick et al., 2017), a noteworthy representative of such methods, as an example, $\mathcal{L}^t$, the loss at task $t$ can be computed by the task loss and an additional term for regularizing each parameter:

$$\mathcal{L}^t = \ell^t(\theta) + \sum_{t'=1}^{t-1} \sum_i \frac{\lambda}{2} F_i (\theta_i - \theta_{t',i}^*)^2,$$

where $\theta$ is the model parameter and $F_i$ is the Fisher information matrix at each parameter $i$.

The regularization itself enables the stability of the entire network. In order to improve the plasticity, we attempt to control the backbone's weight only so that the adapters remain unregularized and thus are able to move freely for parameter exploration. Hence, the integration of adapters to such methods is achieved by ruling out the adapter parameters from regularization, i.e. $i \notin \theta_a$:

$$\mathcal{L}^t = \ell^t(\theta) + \sum_{t'=1}^{t-1} \sum_{i \notin \theta_a} \frac{\lambda}{2} F_i (\theta_i - \theta_{t',i}^*)^2.$$

This modification is applicable to all the weight-regularized methods, as long as they involve parameter-level consolidation. Methods such as Memory Aware Synapses (MAS) (Aljundi et al., 2018) and Path Integral (Path Integral) (Zenke et al., 2017) fall into this category and thus apply the above-mentioned adjustments to align with adapters.

## 4 EXPERIMENTS

In this section, we compare our framework on various methods to their counterparts that are not using adapters. We investigate impact of adapters on different settings, including different method types, task scales, task orderings and datasets.

### 4.1 EXPERIMENTAL SETUP

**Datasets** We compare the peformance of different methods on two datasets: CIFAR-100 (Krizhevsky et al., 2009) and ImageNet (Russakovsky et al., 2015). CIFAR-100 consists of images with small resolutions (input sizes of $32 \times 32 \times 3$) and serves as our primary focus for studying the impact of adapters across different settings. We also include ImageNet, which offers more diverse training images at higher resolutions ($224 \times 224 \times 3$). To mitigate training time and resource constraints, we limit our analysis to the first 100 classes from ImageNet. Dataset statistics are summarized in Appendix A.1.

To ensure a fair comparison, we perform hyperparameter tuning and learning rate selection on a validation set (Masana et al., 2022). The validation set is a class-balanced split, compromising $10\%$ samples from the original training dataset, while the remaining $90\%$ serves as our training dataset. Details on the hyperparameter tuning and learning rate selection can be found in Appendix A.2, and the selection of adapter-specific hyperparameters is explained in Section 4.2.

**Network architectures** Following (De Lange et al., 2022; Hou et al., 2019), we employ two different models for the two datasets. For CIFAR-100, which contains small-resolution images, we use ResNet-34. For ImageNet, with larger-resolution images, we use ResNet-18, as suggested in (He et al., 2016).

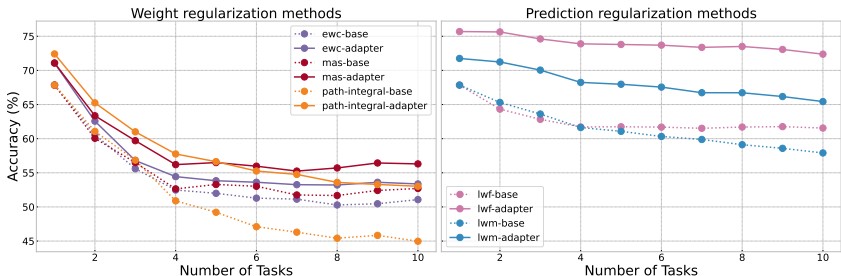

Figure 3: The average accuracy for regularization-based methods with or without adapters on CIFAR-100 (alphabetical ordering) in task-IL. The solid line represents the results with adapter, while the dashed line represents the results without adapter. The left figure displays the performance with weight regularization (EWC, MAS, and Path Integral), and the right figure displays the performance with prediction regularization (LwF and LwM).

**Evaluation metrics**   Two evaluation protocols are commonly used in incremental learning: task incremental learning (task-IL) and class incremental learning (class-IL). Task-IL evaluates the network in a multi-head setting, utilizing a task-ID oracle to determine the appropriate task-specific head at the inference time. In task-IL, the model does not need to differentiate between classes from different tasks. In contrast, class-IL presents a more practical yet challenging scenario where the model must make predictions across all learned classes within a single-head configuration. This requires the model to resolve confusion arising from different tasks. In this section, we focus on task-IL with task-ID information at the inference time , while results for class-IL are included in Appendix B.

To compare the overall learning process across different methods, we use the average accuracy at each task $t$, denoted by $A_t = \frac{1}{t} \sum_{i=1}^{T} a_{t,i}$, and $a_{t,k}$ is the accuracy evaluated on task $k$ after training on task $t$. To ensure reliable and consistent results, we report the averaged results over 10 runs with different random seeds for both CIFAR-100 and ImageNet.

## 4.2    EXPERIMENTAL RESULTS

**On regularization-based methods**   In this section, we study the effect of adapters combined with various weight-regularized methods in incremental learning, including Elastic Weight Consolidatio EWC (Kirkpatrick et al., 2017), Memory Aware Synapses (MAS) (Aljundi et al., 2018), Path Integral (PathInt) (Zenke et al., 2017), as well as prediction-regularized methods such as Learning without Forgetting (LwF) (Li & Hoiem, 2017) and Learning without Memorizing (LwM) (Dhar et al., 2019).

Figure 3 compares different methods with and without adapters in task-IL on CIFAR-100. From the first task onward, weight-regularized methods with adapters exhibit an approximate 3% increase in average accuracy compared to those without adapters. This improvement is consistently maintained throughout the learning process across all methods. For prediction-regularized methods, the accuracy advantage further escalates to as much as 5%. The observed increase in accuracy can be primarily attributed to the model's improved plasticity to learn new task-specific knowledge through the use of adapters, while the backbone was continuously trained as well.

**On task scale**   This paragraph examines the effect of the task scale. The benefits of utilizing adapters diminish as the number of classes increases within each task. As illustrated in Figure 4, when learning either 5 or 10 classes simultaneously, our adapter-based approach continues to effectively capture inter-task differences and significantly outperforms methods without adapters. However, while the advantages are still present, they become less pronounced as the number of classes per task increases. This reduction in the performance margin is understandable, as learning more classes per task not only provides more data for the model to learn but also requires more memory and storage, with fewer regime shifts. In the extreme case, when the number of classes per task reaches 100, incremental learning reduces to a multi-task learning problem.

**On task ordering**   As discussed in Section 3.1 and by Masana et al. (2020), the impact of task orderings on the performance of incremental learning models is often overlooked. While regularization-

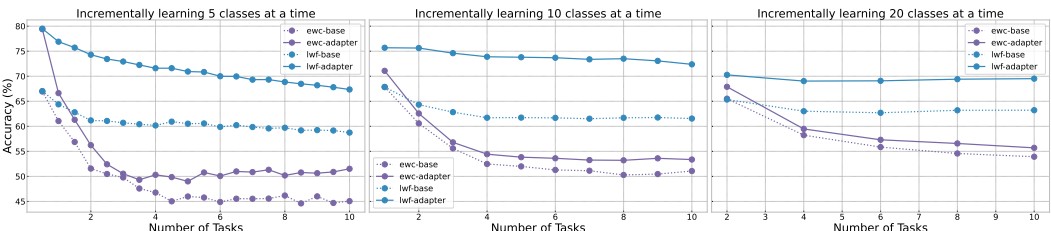

Figure 4: The average accuracy for EWC and LwF with learning 5, 10, and 20 classes at a time on CIFAR-100 (alphabetical ordering) in task-IL.

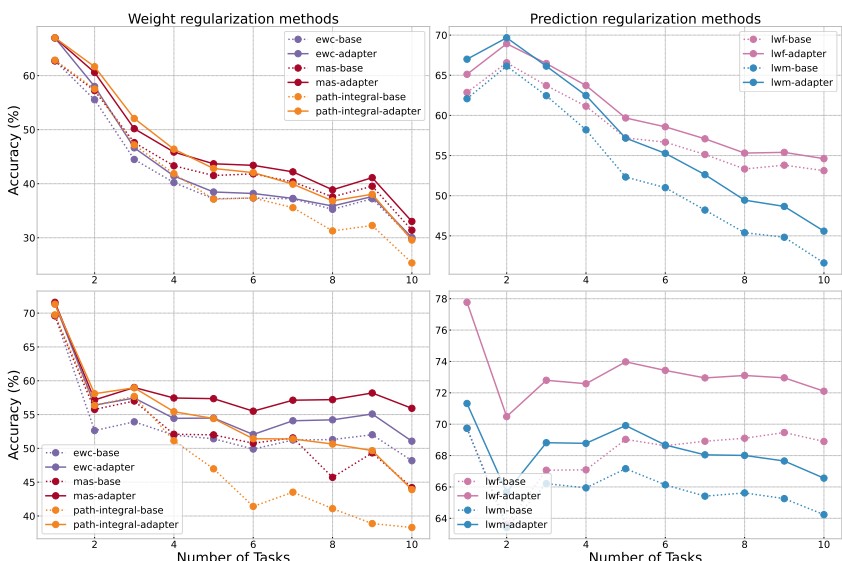

Figure 5: The average accuracy for regularization-based methods with or without adapters on different orderings of CIFAR-100 in task-IL. The upper two figures present the experimental results on the coarse ordering, and the lower two are on the iCaRL ordering. The solid line represents the results with adapter, while the dashed line represents the results without adapter.

based methods suffer from the stability-plasticity dilemma, the introduction of adapters improves both stability and plasticity. This resolution of the stability-plasticity dilemma is attributed to the fact that task-specific knowledge is effectively captured by the adapters, while the backbone continuously learns invariant knowledge. Since all prior experiments discussed in this section were conducted using the CIFAR-100 alphabetical ordering, we have further evaluated all methods using additional orderings, specifically the coarse-grained ordering and the iCaRL ordering.

As shown in Figure 5, methods with adapters are indeed influenced by the varying difficulties associated with different orderings. Although the advantage persists, it diminishes to approximately 1% in some cases. Nonetheless, a general superiority remains evident, as most methods maintain a 3% margin over their non-adapter counterparts across all orderings. In every ordering scenario, adapters consistently exhibit the best overall performance in incremental learning.

**On Imagenet**    This section evaluates our method's performance across larger domain shifts by assessing its performance on ImageNet-Subset, a significantly larger dataset compared to CIFAR-100.

Our method faces certain limitations when applied to ImageNet, as selecting adapter hyperparameters becomes prohibitively expensive on such a large dataset with an average of 10 seeds, which led us to apply the CIFAR-100 hyperparameter setting directly to ImageNet. Additionally, the use of task-specific adapters slows down the generalization process compared to their non-adaptor counterparts. This slowdown occurs because non-adapter methods do not distinguish between task-sharing and

| Method | Task 2 | Task 3 | Task 4 | Task 5 | Task 6 | Task 7 | Task 8 | Task 9 | Task 10 |
|---|---|---|---|---|---|---|---|---|---|
| MAS | 80.4 | 73.6 | 74.4 | 71.3 | 72.8 | 72.9 | 73.5 | 72.1 | 72.7 |
| EWC | 80.3 | 74.6 | 72.0 | 67.8 | 63.2 | 63.9 | 63.6 | 61.4 | 60.8 |
| PathInt | 53.9 | 38.5 | 33.7 | 30.4 | 28.7 | 29.0 | 28.9 | 28.2 | 27.1 |
| LwF | 82.6 | 77.7 | 76.8 | 75.2 | 73.9 | 73.7 | 72.3 | 70.0 | 68.2 |
| LwM | 81.8 | 76.3 | 74.3 | 70.9 | 68.4 | 66.2 | 64.0 | 60.3 | 58.0 |
| MAS-A | 80.0 | 73.6 | 74.0 | 72.2 | 73.3 | **74.6** | **75.0** | **74.2** | **74.2** |
| EWC-A | 76.0 | 67.7 | 68.0 | 67.3 | 67.2 | 68.3 | 67.3 | 65.7 | 65.3 |
| PathInt-A | 76.9 | 68.3 | 67.3 | 65.4 | 65.5 | 67.1 | 67.1 | 65.0 | 65.0 |
| LwF-A | **83.8** | **79.8** | **78.3** | **76.2** | **74.2** | 73.0 | 71.6 | 69.0 | 67.2 |
| LwM-A | 82.8 | 75.9 | 73.9 | 70.6 | 67.8 | 65.9 | 63.2 | 59.4 | 56.9 |

Table 1: The average accuracy for regularization-based methods with or without adapters on ImageNet subset in task-IL. Methods without the "-A" suffix represent the baseline, while those with the suffix include adapters. Following experiments conducted on CIFAR-100 in Section 4.2, these adapters are configured with bottleneck width 128.

task-specific patterns and thus are less impacted on ImageNet, which we only run for 50 epochs as mentioned in Appendix A.2.

While we recommend a more comprehensive study involving careful hyperparameter selection and extended training epochs, our current experimental results, as shown in Table 1, indicate that methods with adapters yield the best performance across all incremental tasks. Even with the hyperparameters from Section 4.2, which are tuned using CIFAR-100 and may not be optimal, methods with adapters still demonstrate non-trivial performance imoprovement.

**On modern IL methods** There is a growing body of incremental learning methods that incorporate task-specific components. We conducted experiments to address two key questions: 1) Can adapters be integrated with these methods, and does such integration further improve performance? 2) Does our adapter integration and training paradigm outperform existing adapter-based methods? For the first question, we integrate adapters with DualPrompt (Wang et al., 2022) and iTAML (Rajasegaran et al., 2020). For the second, we directly compare our approach with TAMiL (Bhat et al., 2023) by aligning our setup with theirs. The experiment details can be found in Appendix A. The results, shown in Table 2, indicate that integrating adapters boosts the original frameworks' performance by more than 1%, and our method outperforms TAMiL, a counterpart that uses attention modules.

## 4.3 ABLATION STUDIES

**The bottleneck width choice** The choice of adapter bottleneck width is crucial to the model performance. We selected EWC (Kirkpatrick et al., 2017) and LwF (Li & Hoiem, 2017) as baseline methods, due to their strong performance among weight-regularized and prediction-regularized methods, respectively. As illustrated in Figure 6, adapters with a bottleneck width of 256 consistently ranked among the top configurations.

**Training with backbone frozen** Various works propose adapter-like network architectures where the backbone is frozen (Liang & Li, 2024; Zhang et al., 2020a). While our framework adapts the backbone to capture task-invariant information and the adapters to capture task-specific information, we hypothesize that freezing the backbone does not support incremental learning. This is because the backbone still requires updates with new, task-inspecific knowledge, which may conflict with the prior knowledge acquired during pre-training. To investigate this, we conduct experiments using the LwF method, where both models are trained with adapters, but one freezes the backbone. Our results, shown in Table 2, demonstrate that the co-trained model, which does not freeze the backbone, outperforms the counterpart. This supports our hypothesis regarding the impact of freezing the backbone.

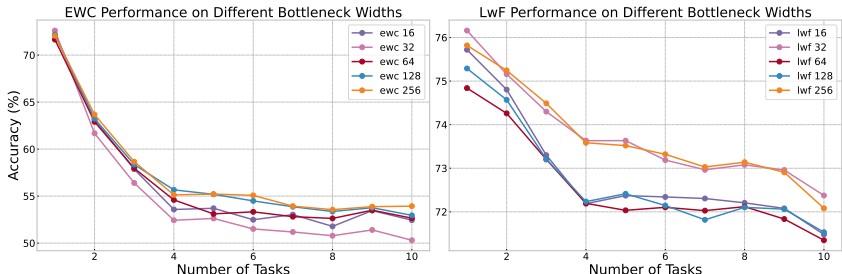

Figure 6: The performance of EWC and LwF methods with different adapter bottleneck width choice on the CIFAR-100 dataset (alphabetical ordering) in task-IL. The suffix `16/32/64/128/256` indicates the method implemented with width 16, 32, 64, 128, and 256, respectively.

| Methods | Acc. | Methods | Acc. | Methods | Acc. | Methods | Acc. |
|---|---|---|---|---|---|---|---|
| DualNet | 88.2 | iTAML | 79.0 | TAMiL | 71.4 | LwF-A | **74.0** |
| DualNet-A | **89.3** | iTAML-A | **80.1** | Adapter+LwF | **74.7** | Lwf-A-FrB | 72.9 |

Table 2: From left to right: DualNet vs. DualNet+adapter, iTAML vs. iTAML+adapter, TAMiL vs. Adapter+LwF (The best method-adapter pair we yielded), and Lwf-A (co-trained) vs. LwF-A (**Fr**ozen **B**ackbone). Test conducted on CIFAR-100, task-IL, top-1 accuracy averaged with 10 tasks is reported.

## 5 CONCLUSION

In this paper, we propose a network design consisting of two blocks: a backbone network for learning invariant features across all tasks and multiple adapters for modeling task-specific knowledge. The backbone and adapters are co-trained continuously in an incremental learning framework. Our extensive experiments conducted on CIFAR-100 and ImageNet, across various orderings and task scales, show that introducing task-specific adapters consistently improves the performance of all considered methods, and exhibits extensive compatibility with all of them. Consequently, we offer an effective solution to resolve the stability-plasticity dilemma for incremental learning, and we envision that future IL algorithms can be benefited from our work, a simple but effective integration of adatpers.

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

# Appendix

## Table of Contents

## A  IMPLEMENTATION DETAILS

We follow the framework FACIL (Masana et al., 2022), and code is implemented using Pytorch. We apply SGD with momentum set to 0.9 and weight decay set to 0.0002. Gradient clipping is set to 1.0.

### A.1  DATASETS

Table 3 presents a summary of the datasets used in the experiments. We apply data augmentation to both datasets, including padding, cropping, input normalization, and random horizontal flipping. Specifically, for CIFAR-100, we apply a padding of 4 pixels to each side of the image, followed by random cropping to $32 \times 32$ for training purposes and center cropping for testing. For the ImageNet-Subset, we resize the images to $256 \times 256$, then perform random cropping to $224 \times 224$ for training and center cropping for testing.

### A.2  LEARNING RATE SEARCH AND HYPERPARAMETER TUNING

We apply the Continual hyperparameter Framework (De Lange & Tuytelaars, 2021), a common framework to select learning rates (LRs) and tune hyperparameters for methods of incremental learning. The framework incorporates two phase: the Maximal Plasticity Search phase to search LR with fine-tuning on the new task, and the Stability Decay phase to search for the optimal hyperparameters.

For the Maximal Plasticity Search phase, we fine-tune the model on the new task and select the optimal LR to achieve high plasticity. Specifically, we train the model from scratch for the first task and apply LR search on {5e-1, 1e-1, 5e-2}. Starting from the second task, the LR search space is limited to 1e-1, 5e-2, 1e-2, 5e-3, 1e-3}. LR decay is applied, with a decay factor of 3 and patience of 10 epochs. The stopping criteria is either the LR below 1e-4 or 100 epochs have passed.

For the Stability Decay phase, we fix the LR and select the hyperparameter starting from a high value, with a gradual decay to achieve the optimal stability-plasticity trade-off. We start from a high value of hyperparameter because this is close to freezing the network such that old knowledge is preserved, and through gradual decay the model becomes less intransigence and slowly converges towards higher forgetting, which would ultimately corresponds to fine-tuning of the previous phase. Specifically, we reduce the hyperparameter value in half if the method accuracy is below the 95% of the fine-tuning accuracy in the previous LR search phase.

Next, we present the starting values for hyperparameters across methods. Most of the methods discussed in our paper are regularization-based approach, which include a hyperparameter $\lambda$ for the regularizer that controls the trade-off between stability and plasticity. The starting values for $\lambda$ are summarized in Table 5. For other method-specific parameters, we generally follow the corresponding original work. For instance, we fix the temperature parameter as 2 for LwF. For PathInt, the dampling parameter is set to 0.1. For LwM, the attention distillation parameter is set to 1 based on an empirical evaluation in (Masana et al., 2022).

| Datasets | # Train | # Validation | # Test | Input size | Batch size |
|---|---|---|---|---|---|
| CIFAR-100 | 45,000 | 5,000 | 10,000 | $32 \times 32 \times 3$ | 128 |
| ImageNet-Subset | 117,000 | 13,000 | 5,000 | $224 \times 224 \times 3$ | 4 |

Table 3: Summary of datasets used. Both datasets contain 100 classes each.

Table 4: The average accuracy for regularization-based methods with or without adapters on ImageNet subset in class-IL. Methods without the "-A" suffix represent the baseline, while those with the suffix include adapters. These adapters are configured with the same bottleneck width and number of layers as those used in the CIFAR-100 experiments.

| Method | Task 2 | Task 3 | Task 4 | Task 5 | Task 6 | Task 7 | Task 8 | Task 9 | Task 10 |
|---|---|---|---|---|---|---|---|---|---|
| MAS | 68.5 | 52.7 | 47.9 | 40.4 | **39.2** | **35.7** | **34.4** | 31.3 | 30.0 |
| EWC | 68.2 | 53.5 | 45.0 | 36.0 | 29.1 | 26.4 | 24.2 | 21.6 | 19.7 |
| PathInt | 41.5 | 22.7 | 15.9 | 11.8 | 9.5 | 8.0 | 7.2 | 6.3 | 5.3 |
| LwF | 69.7 | 57.1 | 51.0 | 44.3 | 38.6 | 33.1 | 29.7 | 26.4 | 24.4 |
| LwM | 66.1 | 50.9 | 44.7 | 36.2 | 29.3 | 23.3 | 20.2 | 17.3 | 15.0 |
| MAS-A | 66.4 | 51.8 | 46.6 | 40.4 | 38.0 | 35.0 | 34.2 | **31.9** | **30.1** |
| EWC-A | 62.1 | 45.1 | 39.0 | 33.9 | 30.7 | 28.0 | 26.4 | 23.6 | 22.0 |
| PathInt-A | 63.0 | 44.5 | 36.1 | 31.1 | 28.7 | 26.4 | 25.1 | 22.7 | 20.7 |
| LwF-A | 71.5 | **59.7** | **53.4** | **45.2** | 38.4 | 32.1 | 28.5 | 24.6 | 22.0 |
| LwM-A | **68.6** | 50.7 | 43.0 | 34.2 | 26.9 | 22.1 | 19.3 | 16.3 | 13.9 |

| hyperparameter | EWC | MAS | PathInt | LwF | LwF-reg | LwM |
|---|---|---|---|---|---|---|
| $\lambda$ | 10,000 | 400 | 10 | 10 | 5+0.5 | 2 |

Table 5: Summary of hyperparameter to control the stability and plasticity used across regularization-based methods. For the LwF with adapter and regularizations (LwF-reg), the two $\lambda$'s ($\lambda_{distill}, \lambda_{\varphi}$) are set to 5 and 0.5 respectively.

Due to computational cost constraints, we have adjusted the stopping criteria for the ImageNet Subset to 50 epochs, rather than the previously mentioned 100. Additionally, we do not conduct LR searches or tune hyperparameters for this dataset, and fix the LR at 0.01.

### A.3 iTAML, TAMiL, and DualPrompt

We implementd iTAML, TAMiL and DualPrompt with their default set of hyperparameters. Specifically,

- For iTAML, we run a ResNet 18 by 70 epochs with memory size 3000, $\mu = 1, \beta = 2$, and $r = 5$. Learning rate is set to 0.01.
- For DualPrompt, we run a ViT-base-patch16-224 with G-prompt and E-prompt enabled. The model is optimized with an adam optimizer with $lr = 0.03$. 2 epochs are run for each task.
- For TAMiL, We run a ResNet 18 by 50 epochs. The learning rate is set to 0.03, and exemplar size is set to 200. The seed is set to 10.

## B Class-IL results

Figure 7 illustrates the average accuracy of all methods in the setup described in Section **??**, specifically for class incremental learning (Class-IL). Figure 8 demonstrates the results across various task scales, while Figure 9 presents the results based on different orderings. Table 4 provides the results for ImageNet.

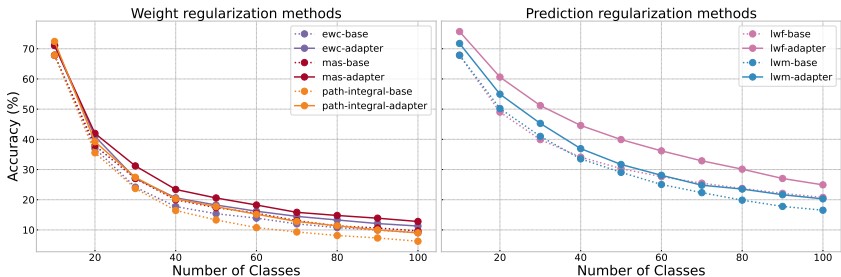

Figure 7: The average accuracy for regularization-based methods with or without adapters on CIFAR-100 (alphabetical ordering) in class-IL. The solid line represents the results with adapter, while the dashed line represents the results without adapter. The left figure displays the performance with weight regularization, and the right figure displays the performance with prediction regularization.

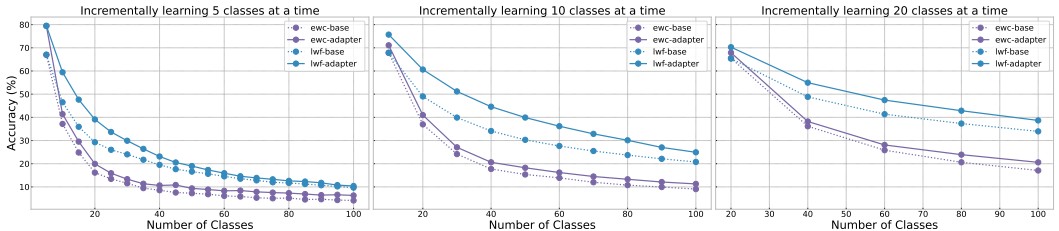

Figure 8: The average accuracy for EWC and LwF with learning 5, 10, and 20 classes at a time on CIFAR-100 (alphabetical ordering) in class-IL.

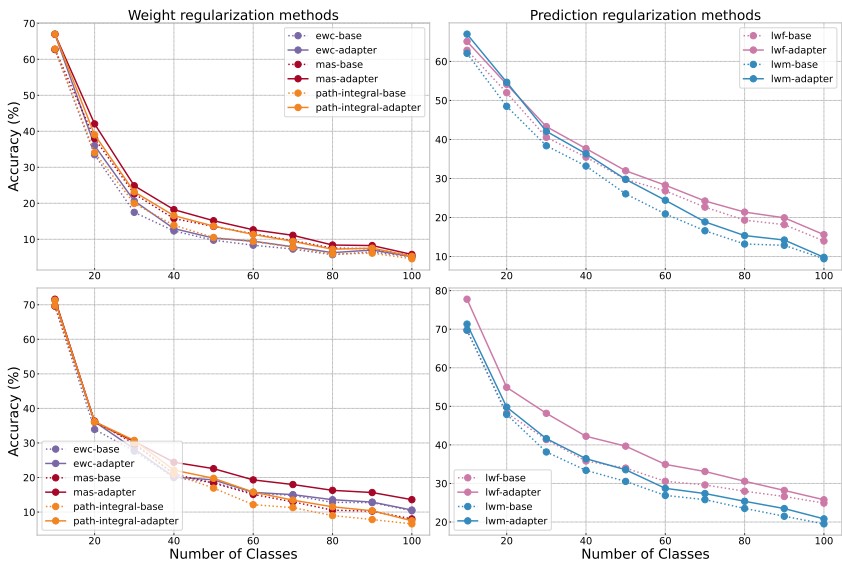

Figure 9: The average accuracy for regularization-based methods with or without adapters on CIFAR-100 (coarse ordering and iCaRL ordering) in class-IL. The solid line represents the results with adapter, while the dashed line represents the results without adapter.

