# OpenReview forum: "Incremental Learning with Task-Specific Adapters"
_ICLR.cc/2025/Conference — Submitted to ICLR 2025_

### Official Review · Reviewer_axyV · 2024-10-21

**Soundness:** 3
**Presentation:** 3
**Contribution:** 1
**Rating:** 3
**Confidence:** 5

**Summary:**

Incremental learning is commonly studied problem in machine learning, seeking models that can learn sequentially arriving data without forgetting prior capabilities. Combatting catastrophic forgetting while maintaining plasticity for new tasks is a common challenge, and the authors propose doing so with task-specific adapters, which are used in conjunction with other continual learning methods like LwF or EWC. Several different formulations of adapters (primarily the number of layers) are introduced, as well as different methods of initialization.  Experiments are performed on CIFAR-100 and ImageNet in both task-incremental and class-incremental learning setups, using ResNet architectures and adding the proposed adapters to several different baseline incremental learning methods.

**Strengths:**

S1. Evaluation protocols: The authors included experiments on both task-incremental learning and class-incremental learning settings. The latter is widely considered to be much harder, with methods that work on task-incremental learning often not generalizing to class-incremental. With the current state of the field, having good results on class-incremental learning is a must for a top-tier conference.

S2. Task order: I also appreciate that the authors analyzed the impact of task order on their method, in particular coarse and iCaRL task ordering instead of the more common default alphabetical ordering. While this effect is not something new from this paper, task order does have a strong impact on results in incremental learning (as confirmed here), and more papers should examine this effect.

S3. Multiple seeds: The authors ran experiments with 10 random seeds, which provides a better sense of consistency.

S4. Writing: The paper is generally clearly written and well-organized. The proposed method and experimental settings are fairly easy to grasp. However, there still remains a number of typos and mistakes. See non-exhaustive examples listed under “Miscellaneous” under Weaknesses. Also, the presentation of the methodology is a little inconsistent with the experiments, as the methodology primarily focuses on LwF, but the experiments present a variety of baselines, including regularization-base approaches; I recommend generalizing the methodology section.

**Weaknesses:**

W1. Novelty: The proposed methodology is lacking in novelty. The adapter strategy (within the larger context of expansion-based methods) for continual learning has been explored many times before (e.g. [a], [b]), and after the recent popularity of LoRA for LLMs, LoRA and similar adaptation techniques have been ported over to continual learning many times as well (e.g. [c], but a simple search will yield many more). [a] in particular is very similar to the proposed method, also using a low-rank parameterization inserted between layers of a network to adapt to each task. All of these highly related methods should be discussed and compared with empirically. Claiming to be “among the first to investigate the role of adapters in incremental learning” (line 085) is not appropriate.

W2. Soundness: From Figure 2, it appears that the main difference between the proposed adapters and the more traditional approach is the lack of freezing of main-body weights, allowing the model the plasticity to adapt to new tasks. However, this freezing is precisely what allows such methods to avoid catastrophic forgetting. Instead, it appears that this method is relying on the “base” continual learning method (e.g. LwF) to prevent most of the forgetting, while the adapters simply provide additional capacity. Also, with the importance given to this distinction, I would expect that this design choice of allowing the main network to be fine-tuned be ablated in the experiments section, but I did not see such a comparison.

W3. Experiments:
- I have concerns about the chosen set of baselines, which are quite old at this point and not representative of the type of method proposed in this paper. There should be comparisons with other adaptation methods/techniques (see W1).
- While there are some gains when added to some baselines in certain settings, the improvements are not clear cut. The paper admits that its improvements are diminished with more classes, or with harder task orderings. For the ImageNet task-incremental learning results in Table 1, the results are even more uneven. While some form of adapter method does the best after every task i, there’s no consistent winner, which poses a problem in practice when a practitioner must choose a single method. Worse, there many instances in Table 1 where adding adapters actually hurts performance: this can be seen for example in EWC and MAS in Task 2, and LWF-A for every task after Task 6, which is especially concerning since LWF-A is highlighted in the paper as one of the main proposed variants.
- While I appreciate the additional class incremental learning results in the Appendix, the gains from adding the proposed adapters appear to be fairly marginal.

W4. Parameter count:
- Any continual learning method that adds parameters per task will raise concerns due to increasing model size + memory requirements as the number of tasks gets large. Experiments in this paper only go up to 10 tasks, which is quite small. Moreover, any sort of characterization of how many additional parameters are being added per task is sorely missing. It’s very important that the authors add the parameter scaling characteristics of the method to the paper.
- Line 226-227: “When applied to large networks, the number of parameters introduced by adapters becomes negligible” <= This is never demonstrated.

Minor:
- Page limit: The page limit for this year’s ICLR has a recommended cap of 9 pages of main text; exceeding this to use a tenth page should only be for including larger and more detailed figures. This paper leaks over to a tenth page, but I do not think it’s justified in doing so. With some judicious editing/rearrangement, this paper should easily fit in 9 pages (e.g. see white space on page 6).
- The paper’s figures tend to be in the middle of the page. While there isn’t anything inherently wrong with that, I’d recommend putting them at the top of pages (or occasionally the bottom); this should help with the space issues mentioned in the previous bullet, while also making the captions more easily distinguishable from the main body text.
- The experimental results section begins with a hyperparameter sweep for bottleneck dimension and number of layers in the adapter. While this perhaps may have been the first thing the authors did chronologically while conducting this study, I would recommend presenting this as an ablation study after the “main” results instead.
- Fig 4+5: I suggest using more distinct shades for differentiating the different methods in this plot. The chosen colors are little to close, which may be difficult for example for readers with color blindness.

Miscellaneous:
- “cosine similarities” => “cosine similarity”
- Equation 1: While fairly self-explanatory, $\ell_t$ and $\Theta$ are never defined.
- Line 292: “intput”
- Line 313: “Two evaluation protocols are available for incremental learning” => Domain incremental learning is also a common evaluation protocol. In the case of this paper, it’s probably fine choosing only to evaluate on task-incremental and class incremental learning, but the authors should instead say that these are the two chosen for this paper’s experiments.
- Line 318: “We focus on the task-IL where task-IDs remain unknown at inference time” => This is in fact class incremental learning setting, and backwards from the definition given over the past few sentences.
- Line 469: backwards double parentheses
- Table 1: This table would be more digestible if the differences between the baseline method and baseline-A were better highlighted, e.g. by showing the delta. I’d also recommend adding a column for the average across all tasks.
- Line 698: A $\lambda$ of 10K for EWC seems a bit excessive. Also, there should be a table number + label here.

[a] Verma et al. “Efficient feature transformations for discriminative and generative continual learning.” CVPR 2021. \
[b] Zhang et al. “Side-Tuning: A Baseline for Network Adaptation via Additive Side Networks.” ECCV 2020. \
[c] Liang et al. “InfLoRA: Interference-Free Low-Rank Adaptation for Continual Learning” CVPR2024


Summary: Overall, while there may be some useful findings here, I find this paper to be too similar to prior work, which aren’t properly discussed or compared with. Taking a further step back, this paper takes the common “baseline + small addition” approach, for which I’ve seen dozens of CL papers; such methods rarely see much adoption, as even if they do lead to improvements, it’s an impossible task trying to combine them with every other such method, for which there often isn’t much synergy.

**Questions:**

Q1: Eq 252: Is $model$ the same as $\phi$, as used in Section 3.2? Or is there another linear projection? Regardless, the notation could be improved to make the connection between the equations on line 258 and 252 (please include equation numbers) more explicit.

Q2: Why is adapter representation differentiation necessary? This isn’t explained well in the methodology. Is this more important when using previous-task alignment instead of random initialization (Section 3.3.2).

Q3: Why do the experiments use a deeper network (ResNet-34) for CIFAR-100, compared with the higher resolution and more complex ImageNet (ResNet-18). This feels a little backwards.

Q4: Figure 4: All methods with adapters noticeably outperform the baseline method (~10%) even after the first task. Why is this case? At 10 classes, there should be no catastrophic forgetting, as there’s only been one task.

Q5: Line 395-6: If it is indeed a capacity issue, then it would seem that using a larger size adapter should lead to improvements. Is this something you’ve tried?

---

> ### Author Response · Authors · 2024-11-23
> **Author's Response**
>
> Dear Reviewer,
>
> Thank you for your concerns and valuable advice. We have updated manuscripts, experiments and results to address your concern.
>
> **Novelty**
> Please refer to the global message for our comparison with other works, particularly those utilizing adapters. We acknowledge that line 085 is inappropriate given previous works, and we will address this in the revised version of the manuscript.
>
> **Soundness and Experiments**
> Please refer to the global message for details on new methods, experiments, and updates to the manuscript.
>
> In addition to the experimental results in the updated manuscript, we provide the full results for freezing the network here:
>
> | Model | Task 1 | Task 2 | Task 3 | Task 4 | Task 5 | Task 6 | Task 7 | Task 8 | Task 9 | Task 10 |
> | :---- | :---- | :---- | :---- | :---- | :---- | :---- | :---- | :---- | :---- | :---- |
> | LwF-A-backbone-co-trained | 75.69 | 75.64 | 74.61 | 73.89 | 73.80 | 73.70 | 73.376 | 73.51 | 73.07 | 72.38 |
> | LwF-A-backbone-fixed | 75.07 | 74.36 | 73.42 | 72.39 | 72.47 | 72.23 | 72.13 | 72.48 | 72.44 | 72.22 |
>
> We believe that allowing the backbone to be co-trained significantly improves performance. Additionally, since LwF-adapter outperforms its non-adapter counterparts, we respectfully disagree with the argument that adapters merely provide additional capacity.
>
> **Parameter Counts**
> Our current structure creates adapters based on a 512-dimensional feature and produces $n$ classes depending on the task, with a bottleneck width of 256\. The total number of parameters added is:
> $512 \times 256\+ 256\times 512 \+ 512 \times n$
>
> For a scenario with $n=10$, this results in approximately 267k additional parameters. This is around 1.2% of the size of a ResNet34 model with 21.8M parameters, and the proportion is even smaller if a ViT backbone is used.
>
> *Note: The original version of this paragraph contained a miscalculation. We incorrectly wrote the total parameters added as $512 \times 128 + 128 \times 512 + 512 \times n$, and asserted that it accounted for approximately 0.1% of the backbone size. This error has been corrected, but we include this clarification here to avoid any misunderstanding in the ongoing discussions between us and the reviewer.*
>
> **Notations and Adapter Representation, and Other Minor Fixes**
> Thank you for your careful review of our manuscript. We have fixed the notations and typos in the new manuscript. The adapter representation constraint has been replaced by a backbone regularization, which we believe is more effective.
>
> **Backbone Selection**
> We follow the FACIL framework \[1\] implementation and adhere to their choice of hyperparameters for better alignment. FACIL uses ResNet-18 for ImageNet and ResNet-32 for CIFAR-100. In our work, we modify the CIFAR-100 backbone to the more commonly used ResNet-34, and that is the only change we made.
>
> **Adapters Outperform the Baseline After the First Task**
> During training, we conducted learning rate selections for each task following the process described in Appendix A.2. While this approach may introduce additional uncertainty to the training procedure, integrating adapters with backbones allows the optimization to adapt to a newer set of hyperparameters, potentially improving model performance. However, we believe such discussions extend beyond the scope of this work.
>
> **ImageNet Results and Capacity Issues**
> We hypothesized that capacity limitations could contribute to the observed disadvantages of our method on ImageNet. Although bottleneck experiments indicate that "bigger is better" does not apply universally to bottleneck width, the significantly larger size of ImageNet compared to CIFAR-100 may require a wider feature bottleneck for optimal performance.
>
> In recent experiments conducted on an 80-epoch LwF model with learning rate tuning, we ruled out capacity issues as the primary cause. Increasing training epochs and optimizing hyperparameters substantially enhanced the model's performance on the ImageNet dataset.
>
> We appreciate your careful review of our manuscript and constructive advice. We are looking forward to your new advice, based on our updated results.
>
> \[1\] Class-incremental learning: survey and performance evaluation on image classification. TPAMI

---

> > ### Comment · Reviewer_axyV · 2024-11-24
> >
> > I thank the authors for the effort they put into their responses, including updating the paper and adding experiments. I read through the other reviews when they were first released, and noticed quite a few common themes, including with the content of my review. I've also read the authors' response to each, as well as the global response.
> >
> > In response to the authors’ comment on “several key works were overlooked “: To be clear, this isn't a case of missing a few references or baselines, which I often find common and relatively understandable. Rather, this paper completely failed to consider an entire (popular) class of solutions for incremental learning. Note that adapters have also commonly been used in image continual learning, not just text. It’s also not just line 085 that was problematic; many parts of the paper considerably overplay the novelty of the proposed method, and the experiments should be focused specifically around comparing with other expansion/adapter-based solutions, as opposed to the current focus on regularization. This paper still needs a considerable re-write on how its contributions are presented, even with the most recent changes.
> >
> > Fixed vs frozen backbone:
> > * I still stand by my comment in W2. Allowing the base network to co-train will lead to catastrophic forgetting of the base network features that allow earlier adapters to maintain performance on previous tasks. Instead, this method is relying on prior regularization methods to combat catastrophic forgetting, and is thus not technically an incremental learning method on its own. I would suggest an experiment using just the proposed adapters, without also pairing it with prior regularization methods; I suspect there will be high catastrophic forgetting. I thus don’t find “we co-train the backbone while others freeze it" to be a compelling contribution or way to distinguish from prior adapter-based incremental learning works. In my opinion, a more correct way of presenting the proposed approach is that the authors have found that adding adapters improves the performance of regularization methods, perhaps by providing added modeling capacity per task and thus reducing the load on the regularization to preserve knowledge.
> > * I thank the authors for adding the comparison between freezing vs co-training the backbone; this ablation is much needed given the paper’s claims. On the other hand, while there does seem to be some improvement from co-training, it seems quite small. I suspect that the frozen approach requires different hyperparameter tuning, due to less weights being trained, and with a more thorough hyperparameter search, this gap would be closed.
> >
> > Parameter counts: Something is not right here. The text says 256 while the math below shows 128. I assume the number is actually 256, given the paper says that worked best, and $512 \times 256 + 256 \times 512 + 512 \times 10 = 267264$, matching the quoted 267K additional parameters. Note however, that there's a missing set of parentheses in the above calculation (unless all tasks share the same weight matrices in the adapter and instead there are only per-task biases, which seems unlikely). Assuming a separate adapter per task, it should actually be $(512 \times 256 + 256 \times 512 + 512) \times n$, which for $n=10$ is actually 2.6M. That means that for a relatively modest 10 tasks, the adapters are in fact ~12% the size of the original network, which is *not* insignificant (note that the rebuttal also miscalculates 267K as 0.1% of 21.8M--even if the adapter parameter count was correct, this should 1.2%, not 0.1%). In contrast, see [a] from my original review, with a more efficient adapter parametrization only 1-3% the size of ResNet-18.

---

> > > ### Author Response · Authors · 2024-11-24
> > >
> > > Dear Reviewer,
> > >
> > > We thank you for your new comments on our updated manuscript, as well as your suggestions and advice. Below is a table of new test results to support our clarifications, which we will refer back to in specific paragraphs.
> > >
> > > | Model      	| Task 1 | Task 2 | Task 3 | Task 4 | Task 5 | Task 6 | Task 7 | Task 8 | Task 9 | Task 10 |
> > > |----------------|--------|--------|--------|--------|--------|--------|--------|--------|--------|---------|
> > > | LwF       	| 67.85  | 64.33  | 62.83  | 61.72  | 61.74  | 61.69  | 61.52  | 61.70  | 61.76  | 61.56   |
> > > | LwF-A-no-reg  | 72.41  | 70.62  | 69.71  | 68.57  | 68.84  | 68.92  | 68.56  | 68.89  | 69.05  | 68.93   |
> > > | LwF-A     	| 75.69  | 75.64  | 74.61  | 73.89  | 73.80  | 73.70  | 73.38  | 73.51  | 73.07  | 72.38   |
> > >
> > > **Method Novelty**
> > > We no longer claim to be the first work to integrate adapters into incremental learning tasks, as our initial research clearly missed relevant prior works. We acknowledge this mistake and thank you and other reviewers for bringing it to our attention. This feedback has strengthened our work by prompting a more comprehensive comparison with current research.
> > >
> > > Regarding novelty, we believe it lies in two aspects:
> > > 1\. By selecting one adapter output rather than integrating multiple modules, we improve the compatibility of adapter-based networks. This is demonstrated by the integration of adapters to various algorithms ranging from EWC to iTAML, and relative performance enhancement.
> > >
> > > 2\. By applying specific regularizations, the performance of adapters can be significantly enhanced. For example, as shown in the second and third rows of the table, task-specific adapters without regularization achieve an average accuracy of 69.5%. This figure improves to 74.0% with the introduction of regularizations.
> > >
> > > We hypothesize that the performance enhancement stems from addressing the stability-plasticity dilemma through a two-step solution:
> > > \- The backbone (feature extractor) controls task-invariant information.
> > > \- Submodules (adapters) control task-specific information.
> > >
> > > By following this guideline, we achieve an efficient design of backbone-adapter modules and novel regularizations. This is validated by significant performance improvements in classic IL algorithms like LwF. To the best of our knowledge, recent algorithms and model architectures do not demonstrate similar insights.
> > >
> > > **Fixing the Backbone**
> > > We report test results for the “correct way” mentioned in your comments. Please refer to the first and second rows of the table above. It is evident that adapters improve performance on their own, with regularization further enhancing results. We are happy to add this finding to the manuscript if you feel necessary.
> > >
> > > The results we provided earlier reflect a comparison between LwF methods with adapters and regularization. Adding a backbone regularization introduces an additional layer for downsampling, which may store task-specific information like an adaptive layer. This could account for the observed small differences. While it demonstrates advantages, a more solid experiment would compare methods using adapters without regularization. These results will be reported in our next update.
> > >
> > > **Parameter Counts**
> > > We apologize for the earlier miscalculation. We have revised our calculation to reflect accurate numbers. In the full setup of methods such as LwF, only the previous adapter is stored, reducing the additional parameter count from $10 \\times 267k$ to $2 \\times 267k$, which constitutes approximately 2.4% of the total parameters.

---

### Official Review · Reviewer_BVJv · 2024-11-01

**Soundness:** 2
**Presentation:** 3
**Contribution:** 2
**Rating:** 3
**Confidence:** 4

**Summary:**

In this work, the authors address the challenge of incremental learning, where a model must continually learn new tasks without forgetting previously learned information. While much of the existing research emphasizes improving model stability to retain earlier task knowledge, this paper argues that inter-task differences are a key contributor to forgetting and proposes a novel network design to address this issue.

**Strengths:**

1. The approach includes two main components: a backbone network that captures invariant features shared across all tasks, and adapters—originally introduced for efficient fine-tuning—that serve as feature modifiers for task-specific information. This design allows the model to learn new tasks while preserving knowledge of prior ones.

2. The authors also provide insights into compression with KD-based incremental learning, highlighting failures and disadvantages through demo experiments.

**Weaknesses:**

However, addressing the following concerns could strengthen the paper:

1. The paper's baselines, EWC and LwF, date back to 2017, which limits the study's relevance, given the advancements in incremental learning benchmarks and methods over recent years. Comparing the proposed method with more recent techniques would enhance its impact.

2. The overall approach is not entirely novel, as similar concepts have been extensively explored, especially in meta-learning-based class-incremental learning or dynamic expansion incremental learning approach, which also trains a generalized model and then adapts it to task-specific scenarios. The authors should discuss differences from this approach in depth, such as in [1] and [2].

3. The overall accuracy of the results needs to be more thoroughly addressed to support the paper's claims. For example, in the class incremental learning (CIL) setting, the proposed method reaches only about 68% accuracy on CIFAR-100 for each 10-class increment, while other methods in [1] achieve up to 78% (as shown in Figure 5). This performance gap limits the generalizability of the method. The authors should clarify whether this accuracy gap is due to the network architecture, network constraints, or training strategy.

[1] "An Incremental Task-Agnostic Meta-learning Approach" [CVPR]

[2] "Tkil: Tangent Kernel Optimization for Class-Balanced Incremental Learning" [ICCVW]

**Questions:**

Experimental Clarity: The experimental setup for class-incremental learning (class-IL) is unclear. Task-incremental learning (task-IL) reveals task information, making it feasible for adapters to adjust to task-specific models. However, class-IL does not disclose this information, and the authors have not clearly described how they address this. Without correct task-specific information, adapting the model may not be feasible.

---

> ### Author Response · Authors · 2024-11-23
> **Author's Response**
>
> Dear Reviewer,
>
> Thank you for your valuable advice. We have updated manuscripts, experiments and results to address your concern.
>
> **Outdated Baselines & Novelty**
> Please refer to our new experiments and clarifications provided in the updated manuscript.
>
> **Overall Accuracy**
>
> We would like to emphasize that our accuracy depends in part on the backbone method used to apply regularizations during training. Our newly introduced regularization, in addition to LwF, positions our approach among the best-performing models. It is also worth noting that our results are averaged over 10 seeds, which prioritizes robustness over peak performance.
>
> Furthermore, a direct comparison shows that we outperform TAMiL under their experimental setup. Detailed results and analysis are included in the updated manuscript.

---

### Official Review · Reviewer_uKUN · 2024-11-03

**Soundness:** 2
**Presentation:** 3
**Contribution:** 1
**Rating:** 5
**Confidence:** 3

**Summary:**

This paper studies the class incremental continual learning problem. To tackle the catastopic forgetting problem, the paper used adopters to add new knowledge to the model and keep the old knowledge without forgetting. The paper is well written and has good amount of experiments to show the benifits use of adoptors for the continual learning problem.

**Strengths:**

the paper is well written and the experiments are suffently done to show the benifits of the adopters for continual learning.

It is nice that the paper evaulates the methods, on Imagenet and higher 224 resolution than 32x32 images on CIFAR100.

figure 5, also shows that this approch works with different sizes for each tasks, such as 5,10 and 20.

**Weaknesses:**

please address the similarities and difference between the following works, they share similar flavors to this work and they are discussed. Specifically, side-tuning is very close to this approch would be nice to compare, if not it should be discussed.
Random path selection for continual learning, NurIPS 2019
Side-Tuning: Network Adaptation via Additive Side Networks, ECCV 2020

also would be nice to show that this approach can work for other architectures like vit. I believe it should work without any problems.

for figure 3, it would be nice to show the width on x axis, to show that there is a optimal middle point where the loss in performance is minimal on average.

this might be outside of this papers scope, but is it necessary to add the adopters for every task? is there any conditions that can be used to makes sure the adopters are initialized only if it is necessary.

**Questions:**

please look at my strengths and weakness sections, and if you can adress the weakness section, i am happy to change my ratings.

---

> ### Author Response · Authors · 2024-11-23
> **Author's Response**
>
> Dear Reviewer,
>
> Thank you for your valuable advice. We have updated manuscripts, experiments and results to address your concern.
>
> **Similarities with Previous Works**
> Please refer to the global message for our detailed response to this issue.
>
> **Applicability to Other Architectures (e.g., ViT)**
>
> Yes, this approach is applicable to other architectures. For instance, ViT can be used as a feature extractor with a hidden size of 768\. The only required modification is to adjust the input dimension of the adapters to 768 to match the feature size. Our new experiments, conducted on DualPrompt, utilize a ViT-base-patch16-224 as the backbone, demonstrating the flexibility of our method.
>
> **Conditions for Initializing Adapters**
>
> It is challenging to determine when adapters should be initialized, as no assumptions can be made about the nature of new, incoming data flows.
>
> However, our method is already parameter-efficient compared to alternatives, as adapters consume minimal additional space. For example, TAMiL requires additional attention models, while iTAML stores and searches for the best models for meta-learning.

---

### Official Review · Reviewer_hWAb · 2024-11-03

**Soundness:** 2
**Presentation:** 2
**Contribution:** 1
**Rating:** 3
**Confidence:** 5

**Summary:**

The paper proposes a method for incremental learning by utilizing task-specific adapters in a neural network architecture. This approach is designed to mitigate catastrophic forgetting, a common issue in incremental learning, by combining invariant feature extraction with task-specific modifications introduced through adapters. The authors implement this adapter-based framework on mainly two incremental learning algorithms (i.e.  EWC and LwF), and report improved performance on CIFAR-100 and ImageNet datasets.

**Strengths:**

The paper aims to address a critical issue in incremental learning: the stability-plasticity dilemma. Task-specific adapters, in this context, are a promising approach to balancing stability and plasticity.

**Weaknesses:**

- I am not convinced by the argument that "existing literature primarily focuses on enhancing model stability to prevent catastrophic forgetting of earlier tasks, often overlooking the challenges posed by inter-task differences." There are several methods in the literature that address the stability-plasticity dilemma using task-specific and task-agnostic components, such as [2-3]. I recommend revising the abstract to focus more specifically on the paper's unique contributions.
- Recent adapter-based methods, such as TAMiL [2], DualNet [3], L2P [4], DualPrompt [5], InfLoRA [6], which are tailored for similar continual learning scenarios, could provide a stronger and more relevant comparison.
- Some sections, such as the descriptions of evaluation metrics and adapter initialization, could be condensed or included in an appendix.
- Novelty is questionable, given the incremental contributions relative to existing approaches.
- Although simplicity can be valuable, a thorough analysis is essential to demonstrate whether this proposal performs on par with or better than competing methods.

- Related Works:
     - It is unclear how multi-task learning is directly relevant to this method.
     - Many of the cited works are from before 2020. Please consider referencing more recent studies to support your points and provide a stronger baseline comparison.
     - For parameter-efficient fine-tuning (PEFT) and adapter-based methods, please compare your work with approaches mentioned in [1] and methods applied to CNNs [2].
- The main problem outlined in Line 129 should be reiterated throughout the paper to maintain focus.
- The claim regarding LwF’s limited adaptability due to increased inter-task diversity (Line 158) is unclear. Please explain how this inference was made.
- Methods Section:
     - The explanation of how adapters are co-trained with the entire network is unclear. How is forgetting in the backbone managed, if at all?
     - If there is no such strategy, how does this differ from methods like [2] that also apply task-specific modules in CNNs?
     - Consider reducing emphasis on the adapter design, as it seems straightforward. Instead, provide more detail on how adapters are co-trained, their effect on overall training, and why they are expected to capture task-specific information.

- The choice of baselines, specifically EWC and LwF, may be insufficient, as these methods are now somewhat dated. The absence of recent incremental learning methods reduces the comparative strength of the paper.
     - The statement in line 328-330 regarding the choice of EWC and LwF as baselines "... owing to their superior performance among ..." is debatable and may not hold up with recent literature.
- The method uses two hyperparameters for regularization losses, but only one is reported in Table 3, with no specification for different datasets. This table also shows a high sensitivity of the methods to hyperparameter choices.
- The rank of the adapter, another hyperparameter, exhibits considerable sensitivity to the dataset (and possibly to the architecture, such as ResNet-18 vs ResNet-34), which should be addressed​.
- The authors use a higher-capacity CNN for CIFAR-100 and a smaller version for ImageNet. This is unusual, as recent literature typically applies the same model across datasets. This choice could impact the results and comparisons and should be justified.
- There is no explanation on how task IDs were inferred at inference when these IDs are not given.
- In Figures 5 and 8, the titles of the subplots are incorrect, as they refer to tasks instead of classes, which could mislead readers.
- In multiple instances, the adapter worsens the results, such as in Figure 6 (for EWC and MAS) and Table 1 (for LwF and LwM), with similar patterns in Table 4, Figure 7, and Figure 8 (middle plot). These results indicate that adapters may not consistently enhance performance and may even hinder it in certain cases.
- The results for Class-IL are generally less convincing, as performance is either very marginal in most cases or worsens when using an adaptor compared to Task-IL. This undermines the proposed method’s claim of broader applicability.




**Ref:**

[1] Parameter-Efficient Fine-Tuning for Continual Learning: A Neural Tangent Kernel Perspective. arXiv 2024.

[2] Task-Aware Information Routing from Common Representation Space in Lifelong Learning. ICLR 2022.

[3] Dualnet: Continual learning, fast and slow. NeurIPS 2021.

[4] Learning to prompt for continual learning. CVPR 2022.

[5] Dualprompt: Complementary prompting for rehearsal-free continual learning. ECCV 2022.

[6] InfLoRA: Interference-Free Low-Rank Adaptation for Continual Learning. CVPR 2024.

**Questions:**

in addition to the weaknesses mentioned above:

- Could you clarify what is meant by “feature extraction (Line 38)”? This could improve the overall clarity of the text.
- The claims made in Introduction (Line 58) are only applicable to regularization and distillation-based methods. Please specify this explicitly and make the scope clear in both the abstract and introduction.

---

> ### Author Response · Authors · 2024-11-23
> **Author's Response**
>
> Dear Reviewer,
>
> Thank you for your valuable advice and concerns. We have updated manuscripts, experiments and results to address your concern.
>
> **Questions Regarding Existing Literature**
> We acknowledge this as a shortcoming in the original manuscript. We have updated the manuscript and included new experiments with additional baselines. Please refer to the global message for more details.
> Our comparisons with TAMiL, DualPrompt, and iTAML demonstrate that our method, or integrating our method, consistently outperforms these baselines. Regarding models like InfLoRA, we argue that freezing the backbone does not enhance overall performance, which we have also demonstrated in the updated experiments.
>
> **Hyperparameter Sensitivity**
> We recommend a hyperparameter selection process for each dataset, though it is not mandatory. For all datasets, we use a universal hyperparameter set, and we have updated Table 3 to reflect the previously omitted hyperparameter.
>
> We found that all adapter-based methods outperform their non-adapter counterparts. Our new results on ImageNet highlight that sufficient training epochs are critical for training adapters, rather than hyperparameter tuning.
>
> **Figure Titles**
> We appreciate your advice and have corrected the figure titles accordingly.
>
> **Adapters Hinder Baseline Performance**
> We encourage you to review the updated manuscript. We believe the advantage of our method remains consistent. For the ImageNet results, we are conducting a more comprehensive study. However, the new results included in the global message already suggest that a significant performance margin will be observed after the updates.
>
> **Class-IL**
> Please refer to the updated manuscript. Although the margins in Class-IL improvements are small, the enhancements remain consistent across the board.

---

### Official Review · Reviewer_sSBW · 2024-11-03

**Soundness:** 1
**Presentation:** 1
**Contribution:** 1
**Rating:** 1
**Confidence:** 5

**Summary:**

This paper tries to access continual learning with the perspective on the 'plasticity' instead of 'stability', which is under explored perspective in the regularisation based methods. To increase the plasticity while minimising the memory consumption required by the proposed method, the authors incorporate the Parameter Efficient Finetuning (PEFT) modules into the incremental learning. The authors introduce the co-training for dividing the role of back bone and PEFT module. The authors validate the proposed methods on diverse datasets.

**Strengths:**

1. The paper is well organised and easy to follow.
2. The authors tried to visualise the results.

**Weaknesses:**

1. Lack of literature search/comparison with existing methods. The fields of continual learning is now gathering a lot of attention in ML community, which means there are plenty of literatures which deals with various perspective. While authors states that the role of plasticity is under-estimated, it is definitely not true. The community tried to increase the plasticity with various approach including architectural expansion or using PEFT modules. There are so many such papers so that I can not even cite them here. However, authors did not cite, discuss or compare them at all which makes me hard to evaluate the contribution of this paper. While I believe that regularisation approach is no more mainstream of continual learning research, the authors did not even compare with the current regularisation approaches. The methods they compare are EWC, LwF things which are far outdated. If the authors conducted some kinds of theoretical research, I might be convinced with these kind of experimental design. However, what authors are doing is proposing practical component for continual learning. In this case, the authors should compare with at least 'recent' (maybe not state of the art) methods.

2. Lack of novelty/Naive approach. As I mentioned in 1, there are plenty of methods that incorporates PEFT modules for Continual learning. Hence, naively integrating with small modification does not have contribution. Also the authors should clarify how the modules are used during inference. According to adapter regularisation, it seems that the authors retain all modules trained so far, however it is not clear how they use the modules for inference, e.g., how they select or integrate.

**Questions:**

See the weakness section

---

> ### Author Response · Authors · 2024-11-23
> **Author's Response**
>
> Dear Reviewer,
>
> Please refer to our updated manuscript and the global message for details on the revisions. We acknowledge that our original submission lacked sufficient references to recent adapter-based and task-specific works. In the revised version, we have included, compared, and discussed these works, highlighting their differences from our approach and their performance in relation to ours.
>
> We look forward to hearing more detailed feedback on our paper and addressing any additional concerns you may have. Thank you for your advice.

---

### Author Response · Authors · 2024-11-23
**Global Message**

Dear Reviewers,

Thank you for your valuable suggestions. We have made several updates to our paper, and we outline the major changes below.

### Reference to More Recent Baselines

We acknowledge that several key works were overlooked in our original manuscript. This was an oversight, and we have addressed it in the updated version. Below, we outline how our approach differs from these works.

Most prior works differ from our approach in three main ways:

1. Some task-specific works do not utilize the capability of adapters, despite using task-specific approaches \[6\]\[7\].
2. Some methods freeze the backbone and train only a small subnet \[2\]\[4\]. In contrast, we co-train the backbone with adapters, allowing them to capture task-invariant and task-specific information, respectively. The updated manuscript demonstrates that this co-training approach improves performance.
3. Certain methods introduce custom losses or combine adapter outputs, which can reduce compatibility with new algorithms \[1\]\[3\]. In our work, we only train and update the adapter created for the current task. Whether to use the old adapter outputs depend on specific algorithmic designs.

Our approach is compatible with all prior baselines, supports customized regularizations, and adapts to new methods. To strengthen our findings, we conducted additional experiments integrating adapters into DualPrompt (L2P) and iTAML. In both cases, we observed improved performance with the use of adapters.

### New Experiments

We have conducted additional experiments to address concerns about outdated evaluations. These new experiments on DualPrompt, iTAML, and TAMiL demonstrate clear advantages over non-adapter counterparts (e.g., DualPrompt and iTAML) and similar network architectures (e.g., TAMiL). Notably, TAMiL requires 200 exemplars, whereas our approach requires none.

We also re-evaluated the adapter regularization posed on LwF. Upon reflection, we found it to be a duplication of the distillation loss and thus not impactful. Instead, we developed a new regularization strategy. This approach restricts backbones to be more similar (without forcing them to be identical) rather than diversifying adapters. Updated experiments with this new regularization show an average performance improvement of 5%. These findings have been incorporated into the revised manuscript.

Concerns were raised regarding the soundness of our ImageNet results. In the original manuscript, we noted that our performance was constrained by limited training epochs due to computational resource limitations. We have since updated our ImageNet training for the LwF-Adapter baseline, with the following results:

| Task Number | 0 | 1 | 2 | 3 | 4 | 5 | 6 | 7 | 8 | 9 |
| :---- | :---- | :---- | :---- | :---- | :---- | :---- | :---- | :---- | :---- | :---- |
| LwF | 85.3 | 84.8 | 81.8 | 80.0 | 78.0 | 76.5 | 75.6 | 74.9 | 72.9 | 70.8 |
| LwF-A | 89.2 | 88.9 | 86.2 | 86.0 | 85.3 | 85.2 | 85.7 | 85.5 | 84.9 | 84.8 |

With more comprehensive training and learning rate tuning, our approach significantly outperforms the original LwF baseline. We anticipate completing the remaining experiments and will include these updates in the camera-ready version.

### Updated Manuscript

We have incorporated all the aforementioned findings and experimental results into the updated manuscript. Notably, we 1\. removed sections about multi-layer adapters and adapter initialization, as by extensive ablation studies we do not find them effective, 2\. added sections for a new regularization technique posed on LwF, and updated experimental results, and 3\. conducted new experiments on more recent baselines and reported the results.

\[1\] Dualnet: Continual learning, fast and slow. NeurIPS 2021\.
\[2\] Side-Tuning: A Baseline for Network Adaptation via Additive Side Networks.  ECCV 2020\.
\[3\] Task-Aware Information Routing from Common Representation Space in Lifelong Learning. ICLR 2022\.
\[4\] InfLoRA: Interference-Free Low-Rank Adaptation for Continual Learning. CVPR 2024\.
\[5\] An Incremental Task-Agnostic Meta-learning Approach
\[6\] DualPrompt: Complementary prompting for rehearsal-free continual learning

---

### Meta-Review · Area_Chair_u39z · 2024-12-19

**Metareview:**

The manuscript received ratings of 3, 5, 3, 1, and 3. Reviewers rasied several concerns with the manuscript including, lack of comparison with existing methods, limited novelty, missing ablations, generalizability analysis with respect to other architectures such as ViT, analysis with regard to initialization of adopters, missing details, performance gaps, and analysis with respect to overall accuracy of the results (e.g., accuracy on CIFAR-100 for each 10-class increment), parameter count, and unclear experimental setup for class-incremental learning. Authors provided a rebuttal to address the concerns of reviewers. Post-rebuttal, while the reviewers appreciated the rebuttal they expressed that their concerns are not fully addressed and therefore remained negative with their overall assessment. One reviewer pointed out that the manuscript "still needs a considerable re-write on how its contributions are presented, even with the most recent changes". Given the reviewers comments, rebuttal and discussions, the recommendation is reject.

**Additional Comments On Reviewer Discussion:**

Reviewers rasied several concerns with the manuscript including, lack of comparison with existing methods, limited novelty, missing ablations, generalizability analysis with respect to other architectures such as ViT, analysis with regard to initialization of adopters, missing details, performance gaps, and analysis with respect to overall accuracy of the results (e.g., accuracy on CIFAR-100 for each 10-class increment), parameter count, and unclear experimental setup for class-incremental learning. Post-rebuttal, while the reviewers appreciated the rebuttal they expressed that their concerns are not fully addressed and therefore remained negative with their overall assessment. One reviewer pointed out that the manuscript "still needs a considerable re-write on how its contributions are presented, even with the most recent changes". Reviewers also remained concerns related to parameter count, efficiency and scalability of the proposed method.

---

### Decision · Program_Chairs · 2025-01-22

Reject